# Diet and Lifestyle Factors and Incident Acute Mesenteric Ischemia—A Prospective Cohort Study

**DOI:** 10.3390/nu17010147

**Published:** 2024-12-31

**Authors:** Yasmin Soltanzadeh-Naderi, Stefan Acosta

**Affiliations:** 1Department of Clinical Sciences, Lund University, 21428 Malmö, Sweden; stefan.acosta@med.lu.se; 2Vascular Center, Department of Cardiothoracic and Vascular Surgery, Skåne University Hospital, 20502 Malmö, Sweden

**Keywords:** acute mesenteric ischemia, risk factors, diet, physical activity, alcohol, smoking

## Abstract

Background/Objectives: Acute mesenteric ischemia (AMI) is life-threatening and difficult to diagnose in time. Unlike many cardiovascular diseases, the association between lifestyle factors such as diet, alcohol consumption, and physical activity and AMI is unknown. Methods: This study is a prospective cohort study with 28,098 middle-aged participants with a mean follow-up time of 23.1 years. Baseline characteristics were obtained with questionnaires regarding physical activity, lifestyle, and diet. The primary endpoint was a diagnosis of AMI identified through the Swedish National Patient Register. Follow-up times were decided by the date of diagnosis, death, or end of follow-up, 2022-12-31. Results: The total number of patients with AMI was 140. Current smoking (adjusted hazard ratio [aHR] 3.02, 95% confidence interval [CI] 1.91–4.79) and those with the highest alcohol consumption (aHR 2.53, 95% CI 1.27–5.03) had a higher risk of developing AMI. Participants with high physical activity, 25.1–50.0 metabolic equivalent task hours per week (MET-h/week), had a lower risk (aHR 0.51, 95% CI 0.27–0.95). Diet quality and dietary components did not affect the risk of AMI. Conclusions: Smoking and higher alcohol consumption were associated with higher risk, while physical activity was associated with lower risk of AMI in this prospective cohort. Diet quality and dietary components were less relevant for the prediction of AMI than these traditional risk factors of atherosclerotic disease.

## 1. Introduction

Acute mesenteric ischemia (AMI) is a critical condition characterized by sudden interruption of blood flow to the mesenteric organs. This blockage can lead to extensive bowel ischemia and, if left untreated, rapid progression to intestinal necrosis. The most common reasons for AMI are acute arterial occlusion of mesenteric arteries, especially the superior mesenteric artery (SMA), non-occlusive mesenteric ischemia (NOMI), and mesenteric vein thrombosis [1]. In a recent population-based study from Estonia, 60% of participants had acute SMA occlusion, 7% had inferior mesenteric artery occlusion, 7% had NOMI, 4% had mesenteric venous thrombosis, and 21% had unclear etiology [2]. In another population-based study with an autopsy rate of 29% from Helsinki, Finland, the 90-day mortality rate in acute occlusive AMI was very high, 83% [3]. One of the main reasons for the high mortality rate is delayed diagnosis. One reason for this clinical challenge is the limited understanding of its risk factors. Known contributing factors include atrial fibrillation, hypercoagulability, and underlying cardiovascular disease [1]. However, unlike other thromboembolic diseases such as stroke and myocardial infarction [4], the associations between dietary patterns, physical activity, and other lifestyle factors and AMI are unknown.

The Malmö Diet and Cancer Study (MDCS) [5] cohort is a comprehensive population-based cohort that started in Malmö, Sweden at the beginning of the 1990s. The purpose of the MDCS was to examine the association between diet, lifestyle, and the risk of developing cancer and other chronic diseases such as cardiovascular diseases and diabetes.

The aim of this study is to identify the risks of dietary and lifestyle factors and incident AMI.

## 2. Materials and Methods

### 2.1. Study Population

This prospective cohort study includes participants from the MDCS cohort [5]. Baseline examinations were conducted on adults aged 46–73 in Malmö, Sweden, from 1991 to 1996. The cohort was followed until the end of 2022. Out of the 30,446 participants included, 28,098 participants underwent anthropometric measurements and a dietary assessment, forming the final study population. All subjects gave their written consent for participation in the study.

### 2.2. Lifestyle and Clinical Data

Baseline data on participants’ lifestyle factors were collected through questionnaires covering smoking status, alcohol intake, physical activity level, and dietary habits [5]. Diet quality was based on Swedish nutritional recommendations and the Swedish dietary guidelines [6]. Information on dietary habits was collected through a diet history method based on a seven-day food diary whereby the participants recorded their food intake during lunch and dinner meals and their intake of cold beverages, together with a 168-item food frequency questionnaire whereby the frequency and portion size of foods regularly consumed during the past year were documented [7]. A picture booklet was included in the questionnaire to evaluate portion sizes. A one-hour interview collected detailed data on serving sizes, cooking practices, and recipes of the foods recorded in the diary. The validity of two dietary assessment methods, (I) a combined food record with a quantitative food frequency questionnaire and (II) an extensive food frequency questionnaire, was evaluated to determine which one should be used for collecting dietary data at baseline in the MDCS [8], and the combined food record method used in the present study was found to be a better and reasonably accurate method compared to the reference method (an 18-day weighed food record method) for food assessment. A diet score was established based on six dietary components: Saturated fat, polyunsaturated fat, fish and shellfish, dietary fiber, fruit and vegetables, and sucrose. Individuals who reached the recommendations for each of the dietary components outlined in the Swedish nutrition recommendations and the Swedish dietary guidelines were given one point. For saturated fat, an adjusted cut-off point was used, since only 2% of the study population reached the recommendation levels. The cut-offs for reaching the recommendations were therefore: polyunsaturated fat 5–10 E%, fish and shellfish ≥ 300 g/week, sucrose ≤ 10 E%, dietary fiber ≥ 2.4 g/MJ, saturated fat ≤ 14 E%, and fruit and vegetables (fruit juices excluded) ≥ 400 g/d. The diet score ranged from 0 to 6. The diet score was also categorized as low (0–1 points), medium (2–4 points), and high (5–6 points). Physical activity was quantified in metabolic equivalent task hours per week (MET-h/week), adapted from a standardized questionnaire. Participants were asked to declare an average of minutes spent on 17 physical activities per week in their spare time, which were thereafter converted into MET-h/week using the method presented by Ainsworth et al. [9].

Height, weight, and blood pressure after ten minutes of rest were registered. Hypertension was defined as a systolic blood pressure ≥ 130 mm Hg and/or diastolic blood pressure ≥ 85 mm Hg or if an individual used anti-hypertensive medication. Diabetes mellitus was defined as fasting blood glucose > 6.0 mmol/L, use of antidiabetic drugs, or self-reported physician’s diagnosis.

### 2.3. Endpoint Ascertainment

The primary endpoint was a diagnosis of AMI identified through the Swedish National Patient Register using ICD-9 (557) and 10 (K55.0) codes specific to AMI. Follow-up time was decided by the date of diagnosis, death, or end of the study follow-up period on 31 December 2022. The diagnoses of AMI (K55.0) in the Swedish National Patient Register were validated by selecting a sample of study participants between 1 January 2010 and 31 December 2022 (Table 1). The study participants’ patient records and available radiological images, particularly computed tomography (CT) angiography, were scrutinized. In total, 81 participants were identified, of whom 4 patients were primarily treated in hospitals outside Region Skåne. None of these four patients’ records or images could be retrieved.

### 2.4. Statistics

Baseline characteristics were expressed as the median with interquartile range (IQR) for the continuous variables, and the categorical variables were expressed as the total and percentages. Hazard ratios (HR) with 95% confidence intervals (CI) were calculated using Cox proportional hazard models. Continuous variables were tested for normal distribution with the Kolmogorov–Smirnov test. All skewed variables were log-transformed, and HRs were expressed per 1 standard deviation (SD) increment. All variables in the crude analysis were included in the multivariable analysis. The lowest exposure in each variable was used as the reference group, except for alcohol where quintile 1 was chosen as the baseline since zero consumption has been shown to be associated with increased incidence of cardiovascular disease [10,11]. Statistical analyses were performed using IBM SPSS Statistics 28 (SPSS, Chicago, IL, USA), with significance set at *p* < 0.05.

## 3. Results

### 3.1. Baseline Characteristics

A total of 28,098 participants were included in the analysis, of whom 140 (0.5%) developed AMI during a mean follow-up period of 23.1 years. Compared to those without incident AMI, individuals who developed AMI were older and more likely to be male (Table 2).

### 3.2. Risk Factors for AMI

In the multivariable analysis, several factors were significantly associated with an increased risk of AMI (Table 2). The risk of AMI increased with age with HR of 2.01/one standard deviation increment (95% CI 1.62–2.48). High alcohol intake was associated with a greater risk of AMI. Specifically, participants in the highest quintile of alcohol consumption had an HR of 2.53 (95% CI 1.27–5.03) compared to the reference group. Current smokers had a significantly higher risk of AMI (HR 3.02, 95% CI 1.91–4.79) compared to never-smokers.

### 3.3. Protective Factors for AMI

Participants with higher levels of physical activity had a lower risk of AMI (Table 2). Those engaging in 15.1–25.0 MET-h/week and 25.1–50.0 MET-h/week had both lower risks in the crude analyses (HR 0.51, 95% CI 0.26–0.99 and HR 0.48, 95% CI 0.26–0.89, respectively). When adjusted, only 25.1–50.0 MET-h/week remained significant (HR 0.51, 95% CI 0.27–0.95).

### 3.4. Other Variables

Other factors, such as body mass index (BMI), hypertension, diabetes mellitus, educational level, and diet quality and dietary components, did not show statistically significant associations with the risk of developing AMI in the multivariable model (Table 2).

## 4. Discussion

The present study shows that the risk of incident AMI is significantly higher if one is a current smoker at baseline. Smoking is a well-known risk factor for many cardiovascular diseases in the arterial system, such as stroke and peripheral artery disease, and in venous thromboembolism [12]. Smoking is believed to have a causal effect; however, the precise pathogenesis is unclear. The theory is that smoking, among other things, leads to oxidative stress, which has a central role in atherosclerosis [13]. Notably, the dose–response was unknown in the study cohort. This finding supports the recommendation that medical treatment of AMI should include smoking cessation [1].

Those consuming high amounts of alcohol had an increased risk of incident AMI compared to baseline. Interestingly, the groups with the highest risk increase were quintile 2 (0.9–4.3 g/day for women/3.4–9.1 g/day for men) and quintile 5 (>14.0 g/day for women/>25.7 g/day for men). It is well known that alcohol consumption is associated with an increased risk of cardiovascular diseases such as atrial fibrillation, where it seems to have a linear relationship [14], and in peripheral artery disease, where a meta-analysis showed a possible U-shaped association [10]. It is important to remember that these papers do not study causal relationships but rather associations, and results could be explained by confounding. For example, the zero-consumer group may include a higher proportion of former alcoholics or individuals with underlying somatic or psychiatric illnesses. Furthermore, occasional drinking may be associated with higher socioeconomic status and a more active social life. Thus, caution is warranted before drawing causal conclusions based on these associations.

In the present study, physical activity of 25.1–50.0 MET-h/week (HR 0.51, 95% CI 0.27–0.95) was the only quantile with lowered risk of AMI. The association between physical activity and cardiovascular mortality has previously been studied in this cohort by Bergwall et al. [15], showing a decrease in risk with 15–25 MET-h/week (HR 0.84, 95% CI 0.72–0.97). It is of value to highlight that the number of cases in each quantile in the present study varied between 17–37, a notably lower sample size when compared to the study by Bergwall et al. [15], yet with a stronger association, possibly indicating that there is a strong association masked by a type II error. A meta-analysis looking at the associations between physical activity and cardiovascular diseases found that an increase in MET h/week was associated with a reduction in the risk of cardiovascular mortality, with the highest effect on inactive individuals [16]. The exact reason for physical activity being protective is not known but is believed to be due to a combination of processes such as decreased inflammation, improved endothelial function, and lowered oxidative stress in vascular tissue [17].

The studied dietary components did not have an association with the risk of AMI. Apart from there being a low sample size in the AMI group, increasing the risk of II error, other reasons could be that the dietary components analyzed may be less relevant for AMI. Other relevant variables could be the amount of processed foods or sources of macronutrients [18,19]. Diet quality [6] is based on Swedish National Food Agency [20] and Nordic Nutrition Recommendations [21], which are mostly based on associative findings, which are not the highest grade of evidence. Diet is a complex and unfortunately difficult variable to study, where causal conclusions can very seldom be suggested.

Interestingly, hypertension had no significant association with increased HRs of AMI. Hypertension has previously been shown to be common in patients with AMI, both in retrospective [2] and prospective [22] studies. One reason could be that hypertension has a high prevalence, with 69.8% of individuals having hypertension in the AMI group and 61.5% of individuals having hypertension in the non-AMI group, both rather high numbers. Hypertension was defined with two methods, either self-reported or based on one single value of blood pressure over 160 mm Hg systolic or 95 mm Hg diastolic at inclusion examination [23], with the latter method being the larger error source. Another reason might be that AMI patients in some senses are a heterogenous group regarding its risk factors. Arterial and venous AMI affect different patient categories, with the latter being more associated with younger patients with coagulopathy or malignancies, whereas arterial—both occlusive and in many cases non-occlusive—might be more associated with “classic” cardiovascular risk factors [1].

The detailed prospective data on diet and lifestyle factors collected at baseline and the long follow-up period, a mean of 23.1 years, are major strengths of this study. The validation of a sample of patients with AMI induces scientific rigor, showing that most patients had an arterial AMI. However, this validation showed that a major limitation of the present study was that almost 1/5 of patients were confirmed to not have AMI, and there were also uncertainties about the final diagnosis in an additional 1/5 of the patients. These two relatively large subgroups were also composed of different patient entities as eight out of 14 patients in the former group had milder left-sided colonic ischemia with no need for bowel resection and all survived, whereas 13 out of 15 in the latter group had clinical suspicion of AMI with a mortality rate of 76.9%, possibly indicating that some indeed had AMI. Hence, it is very important to validate and scrutinize register data in patients with AMI.

## 5. Conclusions

In conclusion, smoking and higher alcohol consumption were associated with higher risk, while physical activity was associated with lower risk of AMI in this prospective cohort. Diet quality and dietary components were less relevant for the prediction of AMI than these traditional risk factors for atherosclerotic disease. A primary prevention strategy in society in the management of smoking, alcohol, and physical activity may play a key role in lowering incident AMI.

## Figures and Tables

**Table 1 nutrients-17-00147-t001:** Validation of available participant data registered as K55.0 (acute mesenteric ischemia).

	Number of Participants (*n* = 77)	CT with Intravenous Contrast	Endovascular Therapy	Bowel Resection	In-Hospital Mortality
**Diagnosis confirmed**	48 (62.3)	41 (85.4)	11 (22.9)	16 (33.3)	23 (47.9)
Acute thrombotic occlusion of the SMA (%)	16 (20.8)	15 (93.8)	9 (56.3)	4 (25.0)	6 (37.5)
Acute embolic occlusion of the SMA (%)	10 (13.0)	10 (100.0)	2 (20.0)	4 (40.0)	4 (40.0)
Mesenteric venous thrombosis (%)	6 (7.8)	5 (83.3)	0 (0.0)	0 (0.0)	2 (33.3)
NOMI (%)	13 (16.9)	11 (84.6)	0 (0.0)	7 (53.8)	9 (69.2)
Indeterminate acute arterial mesenteric ischemia (%)	3 (3.9)	0 (0.0)	0 (0.0)	1 (33.3)	2 (66.7)
**Diagnosis confirmed but not primary acute mesenteric ischemia**	14 (18.2)	11 (78.6)	2 (14.3)	3 (21.4)	1 (7.1)
Chronic mesenteric ischemia (%)	3 (3.9)	3 (100.0)	1 (33.3)	0 (0.0)	0 (0.0)
CT- and/or colonoscopy-verified left-sided colonic ischemia (%)	8 (10.4)	5 (75.0)	0 (0.0)	0 (0.0)	0 (0.0)
Mechanical small bowel ileus with intestinal ischemia (%)	2 (2.6)	2 (100.0)	0 (0.0)	2 (100.0)	0 (0.0)
Iatrogenic SMA thrombus after elective endovascular therapy (%)	1 (1.3)	1 (100.0)	1 (100.0)	1 (100.0)	1 (100.0)
**Diagnosis not confirmed**	15 (19.5)	4 (26.7)	0 (0)	3 (20.0)	12 (80.0)
Mechanical colonic ileus due to suspected cancer (%)	1 (1.3)	1 (100.0)	0 (0.0)	0 (0.0)	1 (100.0)
Clinical suspicion of acute mesenteric ischemia only (%)	13 (16.9)	3 (23.1)	0 (0.0)	3 (23.1)	10 (76.9)
Unclear diagnosis (no autopsy) (%)	1 (1.3)	0 (0.0)	0 (0.0)	0 (0.0)	1 (100.0)

SMA; superior mesenteric artery, NOMI; non-occlusive mesenteric ischemia, CT; computed tomography.

**Table 2 nutrients-17-00147-t002:** Baseline characteristics of 28,098 study participants with and without incident AMI in the MDCS cohort.

	Incident Acute Mesenteric Ischemia (*n* = 140)	No Incident Acute Mesenteric Ischemia (*n* = 27,958)	Age-and Sex Adjusted HR (95% CI)	Multivariable * Adjusted HR (95% CI)
**Characteristics**				
Male sex (%)	59 (42.1)	11,004 (39.4)	1.29 (0.90–1.84)	1.18 (0.77–1.82)
Age (years)	61.4 (55.8–65.3)	57.8 (51.3–64.2)	1.82 ^a^ (1.50–2.22)	2.01 ^a^ (1.62–2.48)
BMI (kg/m^2^)	25.7 (23.2–28.7) (*n* = 139)	25.3 (23.0–28.0) (*n* = 27,917)	0.99 ^a^ (0.82–1.20)	1.02 ^a^ (0.84–1.25)
Hypertension (%)	97/139 (69.8)	17,142/27,886 (61.5)	1.29 (0.86–1.93)	1.28 (0.85–1.92)
Diabetes mellitus (%)	7 (5.0)	1223 (4.4)	1.12 (0.46–2.75)	1.17 (0.47–2.89)
**Alcohol Consumption (%)**				
Zero consumers	10 (7.1)	1796 (6.4)	2.26 (0.95–5.37)	2.06 (0.84–5.06)
Quintile 1 (<0.9 g/day for women/ <3.4 g/day for men)	16 (11.4)	5240 (18.7)	1 (Ref)	1 (Ref)
Quintile 2 (0.9–4.3 g/day for women/3.4–9.1 g/day for men)	32 (22.9)	5230 (18.7)	2.32 (1.19–4.53)	2.32 (1.18–4.56)
Quintile 3 (4.4–8.1 g/day for women/9.2–15.7 g/day for men)	22 (15.7)	5241 (18.7)	1.58 (0.77–3.24)	1.66 (0.81–3.40)
Quintile 4 (8.2–14.0 g/day for women/15.7–25.7 g/day for men)	27 (16.6)	5227 (18.7)	1.96 (0.98–3.93)	1.95 (0.97–3.92)
Quintile 5 (>14.0 g/day for women/ >25.7 g/day for men)	33 (23.6)	5224 (18.7)	2.75 (1.40–5.42)	2.53 (1.27–5.03)
**Smoking (%)**				
Never	39 (27.9)	10,605/27,946 (37.9)	1 (Ref)	1 (Ref)
Former	44 (31.4)	9462/27,946 (33.9)	1.47 (0.92–2.35)	1.48 (0.92–2.38)
Current	57 (40.7)	7879/27,946 (28.2)	3.13 (2.01–4.89)	3.02 (1.91–4.79)
**Leisure Time Physical Activity (%)**				
<7.5 MET-h/week	17/138 (12.3)	2715/27,757 (9.8)	1 (Ref)	1 (Ref)
7.5–15.0 MET-h/week	31/138 (22.5)	4129/27,757 (14.9)	1.15 (0.62–2.14)	1.17 (0.62–2.19)
15.1–25.0 MET-h/week	25/138 (18.1)	6384/27,757 (23.0)	0.51 (0.26–0.99)	0.53 (0.27–1.03)
25.1–50.0 MET-h/week	37/138 (26.8)	10,082/27,757 (36.3)	0.48 (0.26–0.89)	0.51 (0.27–0.95)
>50.0 MET-h/week	28/138 (20.3)	4447/27,757 (16.0)	0.85 (0.45–1.61)	0.86 (0.45–1.65)
**Educational Level (%)**				
Less than 9 years	57/138 (51.7)	11,720/27,889 (42.0)	1 (Ref)	1 (Ref)
Elementary school (9–10 years)	43/138 (23.6)	7289/27,889 (26.1)	1.32 (0.86–2.02)	1.39 (0.89–2.14)
Elementary + upper secondary school (9–13 years)	13/138 (8.0)	2478/27,889 (8.9)	1.28 (0.68–2.43)	1.38 (0.72–2.63)
University studies, no degree	13/138 (7.2)	2429/27,889 (8.7)	1.28 (0.68–2.41)	1.38 (0.72–2.62)
University studies, with degree	12/138 (9.5)	3973/27,889 (14.2)	0.79 (0.41–1.53)	0.88 (0.45–1.72)
**Diet Quality**				
Low (%)	28 (20.0)	4266 (15.2)	1 (Ref)	1 (Ref)
Medium (%)	94 (67.1)	19,939 (71.3)	0.66 (0.42–1.04)	0.76 (0.48–1.22)
High (%)	18 (12.9)	3753 (13.4)	0.63 (0.34–1.18)	0.80 (0.42–1.54)
Diet score (0–6)	3 (2–4)	3 (2–4)	0.92 (0.80–1.04)/ point increase	0.97 (0.85–1.11)/point increase
**Dietary Components**				
Total energy intake (kcal/day)	2224.0 (1845.1–2683.5)	2185.1 (1818.4–2631.9)	1.04 ^a^ (0.85–1.27)	1.01 ^a^ (0.82–1.24)
Saturated fat (E%)	15.6 (13.8–18.1)	15.7 (13.6–18.3)	1.08 ^a^ (0.90–1.29)	0.98 ^a^ (0.81–1.19)
Polyunsaturated fat (E%)	5.9 (4.9–6.7)	5.8 (4.8–6.8)	1.05 ^a^ (0.88–1.26)	1.02 ^a^ (0.85–1.22)
Sucrose (E%)	8.4 (5.9–10.9)	8.0 (6.1–10.3)	0.99 ^a^ (0.82–1.20)	1.00 ^a^ (0.83–1.20)
Fiber (g/MJ)	2.1 (1.7–2.4)	2.1 (1.8–2.6)	0.84 ^a^ (0.70–1.01)	0.97 ^a^ (0.80–1.18)
Fruit and vegetables (g/day)	316.7 (232.9–442.4)	346.5 (245.3–474.3)	0.86 ^a^ (0.72–1.02)	0.96 ^a^ (0.79–1.16)
Fish (g/week)	301.3 (108.4–487.4)	277.1 (150.3–440.2)	0.90 ^a^ (0.75–1.07)	0.91 ^a^ (0.76–1.08)

Data are *n* (%) or median (interquartile range, IQR). MDCS; Malmö Diet and Cancer Study, HR; hazard ratio, BMI; body mass index, E; energy, MET; metabolic equivalent of task, MJ; megajoule. ^a^ HRs are expressed per 1 standard deviation increment. * Multivariable models include all risk factors and diet quality respective dietary component variables. Dietary component variables were not mutually adjusted nor with diet quality.

## Data Availability

The datasets analyzed during the current study are not publicly available due to the nature of the sensitive personal data and study materials. However, procedures for sharing data, analytical methods, and study materials for reproducing the results following Swedish legislation can be arranged by contacting the corresponding author or study organization (https://www.malmo-kohorter.lu.se/malmo-kost-cancer-mkc, accessed on 27 December 2024).

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
