# Peer review of "Diet and Lifestyle Factors and Incident Acute Mesenteric Ischemia—A Prospective Cohort Study"

_nutrients, 2024, doi:10.3390/nu17010147_

Round 1
Reviewer 1 Report
Comments and Suggestions for Authors
Very interesting publication and innovative topic. I have few comments, mostly about the methods.
I suggest you make the summary based on the journal guidelines, i.e. add in the appropriate places: "Background/Objectives", "Methods", "Results", and "Conclusions". This will be more legible for the reader.
line 33: "AMI has a high mortality rate [2], " how high? Are there any specific numbers?
In point "2.2. Lifestyle and Clinical Data":
was the questionnaire validated? Did you use, for example, a current food intake questionnaire or a 24-hour interview to assess nutritional components? How many days of information was collected about what the patients ate?
Generally, I do not understand what questionnaires were used to collect the dietary and lifestyle data. Please provide this information.
In Table 1 in the row:
- "Diagnosis confirmed but not primary acute mesenteric ischemia"
- and "Diagnosis not confirmed"
there are unfinished numerical data in the individual groups.
Author Response
Reviewer 1.
Very interesting publication and innovative topic. I have few comments, mostly about the methods.
I suggest you make the summary based on the journal guidelines, i.e. add in the appropriate places: "Background/Objectives", "Methods", "Results", and "Conclusions". This will be more legible for the reader.
Since Nutrients now accepts free format submission, we will insert these suggested subtitles.
line 33: "AMI has a high mortality rate [2], " how high? Are there any specific numbers?
We revised and inserted two new references:
In a recent population-based study from Estonia, 60% had acute SMA occlusion, 7% inferior mesenteric artery occlusion, 7% NOMI, 4% mesenteric venous thrombosis, and 21% unclear etiology [2]. In another population-based study with an autopsy rate of 29% from Helsinki, Finland, the 90-day mortality rate in acute occlusive AMI was very high, 83% [3]. One of the main reasons for the high mortality rate is delayed diagnosis.
New reference 2: Kase K, Blaser AR, Tamme K, Mändul M, Forbes A, Talving P, Murruste M. Epidemiology of acute mesenteric ischemia: A Population-based investigation. World J Surg 2023; 47: 173-181.
New Reference 3: Lemma A, Tolonen M, Vikatmaa P, Mentula P, Kantonen I, But A, Leppäniemi A, Sallinen V. Epidemiology, Diagnostics, and Outcomes of Acute Occlusive Arterial Mesenteric Ischaemia: A Population Based Study. Eur J Vasc Endovasc Surg 2022; 64: 646 – 653.
In point "2.2. Lifestyle and Clinical Data":
was the questionnaire validated? Did you use, for example, a current food intake questionnaire or a 24-hour interview to assess nutritional components? How many days of information was collected about what the patients ate?
We deleted the sentence “For diet assessment, a food frequency questionnaire was used to capture habitual dietary intake.” and expanded this section and inserted new references: “Information on dietary habits was collected through a diet history method based on a seven-day food diary where the participant recorded their food intake at lunch and dinner meals and intake of cold beverages, together with a 168-item food frequency questionnaire where frequency and portion-size of foods regularly consumed during the past year were documented [6]. A picture booklet was included in the questionnaire to evaluate portion sizes. A one-hour interview collected detailed data on serving sizes, cooking practices and recipes of the foods recorded in the diary.” The validity of two dietary assessment methods, (I) a combined food record with a quantitative food frequency questionnaire and (II) an extensive food frequency questionnaire, was evaluated to determine which one should be used for collecting dietary data at baseline in the MDCS [7], and the combined food record method used in the present study was found to be a better and reasonably accurate method compared to the reference method (an 18-day weighed food record method) for food assessment.
New reference 6. Wirfalt, E. et al. A methodological report from the Malmo Diet and Cancer study: development and evaluation of altered routines in dietary data processing. Nutr J 2002 Nov 19:1:3. Doi: 10.1186/1475-2891-1-3.
New reference 7. Elmstahl S, Riboli E, Lindgarde F, Gullberg B, Saracci R. The Malmo Food Study: the relative validity of a modified diet history method and an extensive food frequency questionnaire for measuring food intake. Eur J Clin Nutr 1996;50:143-51.
Generally, I do not understand what questionnaires were used to collect the dietary and lifestyle data. Please provide this information.
By clarifying the dietary data collection, we believe that both dietary and lifestyle data collection including physical activity became clearer.
In Table 1 in the row:
- "Diagnosis confirmed but not primary acute mesenteric ischemia"
- and "Diagnosis not confirmed"
there are unfinished numerical data in the individual groups.
Thank you. We summarized and inserted the data in the respective rows: We add percentages within parentheses for all three rows including the subtitles.
11 (78.6) 2 (14.3) 3 (21.4) 1(7.1)
4 (26.7) 0 (0) 3(20.0) 12 (80.0)
Reviewer 2 Report
Comments and Suggestions for Authors
The work is not groundbreaking, and most of the results it provides are well known, such as the fact that alcohol, smoking or little exercise are risk factors for mesenteric ichemia (as for the vast majority of metabolic diseases). However, other results are not so evident, such as the fact that a diet far from the recommendations does not constitute a risk factor for mesenteric ichemia. The work is based on a very large database of patients, which gives it scientific value.
The number of shelf-ciations is 4/21 (19%). It is too high. The similarity index excluding references in 35%. From them, 10% belongs to the article published by one of the authors entitled “Diet and Lifestyle Factors and Risk of Atherosclerotic Cardiovascular Disease—A Prospective Cohort Study”.
Description of lifestyle and clinical data. Should be more specific. It is not enough to state the score given for example to a diet with a low, medium or high score, but it should also mention on the basis of which specific criteria the authors make this classification.
In general, it is well written and there are no specific issues to be corrected beyond a lack of adaptation of the list of references to the journal standards, which will be corrected in later stages.
Author Response
Reviewer 2.
The work is not groundbreaking, and most of the results it provides are well known, such as the fact that alcohol, smoking or little exercise are risk factors for mesenteric ichemia (as for the vast majority of metabolic diseases). However, other results are not so evident, such as the fact that a diet far from the recommendations does not constitute a risk factor for mesenteric ichemia. The work is based on a very large database of patients, which gives it scientific value.
Thank you
The number of shelf-ciations is 4/21 (19%). It is too high. The similarity index excluding references in 35%. From them, 10% belongs to the article published by one of the authors entitled “Diet and Lifestyle Factors and Risk of Atherosclerotic Cardiovascular Disease—A Prospective Cohort Study”.
We have adjusted this according to the Editors suggestion as well. We deleted one of the four self-citations and now the percentage of self-citations is below 15% (3/24 = 12.5%)
Description of lifestyle and clinical data. Should be more specific. It is not enough to state the score given for example to a diet with a low, medium or high score, but it should also mention on the basis of which specific criteria the authors make this classification.
The diet assessment method has been extensively expanded according to referee 1.
We added: “Information on dietary habits was collected through a diet history method based on a seven-day food diary where the participant recorded their food intake at lunch and dinner meals and intake of cold beverages, together with a 168-item food frequency questionnaire where frequency and portion-size of foods regularly consumed during the past year were documented [6]. A picture booklet was included in the questionnaire to evaluate portion sizes. A one-hour interview collected detailed data on serving sizes, cooking practices and recipes of the foods recorded in the diary.” The validity of two dietary assessment methods, (I) a combined food record with a quantitative food frequency questionnaire and (II) an extensive food frequency questionnaire, was evaluated to determine which one should be used for collecting dietary data at baseline in the MDCS [7], and the combined food record method used in the present study was found to be a better and reasonably accurate method compared to the reference method (an 18-day weighed food record method) for food assessment.
New reference 6. Wirfalt, E. et al. A methodological report from the Malmo Diet and Cancer study: development and evaluation of altered routines in dietary data processing. Nutr J 2002 Nov 19:1:3. Doi: 10.1186/1475-2891-1-3.
New reference 7. Elmstahl S, Riboli E, Lindgarde F, Gullberg B, Saracci R. The Malmo Food Study: the relative validity of a modified diet history method and an extensive food frequency questionnaire for measuring food intake. Eur J Clin Nutr 1996;50:143-51.
The development of the specific criteria for diet score is outlined in reference 5, and this is a complete developmental methodological work of its own.
In general, it is well written and there are no specific issues to be corrected beyond a lack of adaptation of the list of references to the journal standards, which will be corrected in later stages.
According to instructions for authors references can be formatted in various ways, but it should be uniformly written. We use Endnote which is encouraged.
Citation from the Instructions for authors:
Your references may be in any style, provided that you use the consistent formatting throughout. It is essential to include author(s) name(s), journal or book title, article or chapter title (where required), year of publication, volume and issue (where appropriate) and pagination. DOI numbers (Digital Object Identifier) are not mandatory but highly encouraged. The bibliography software package EndNote, Zotero, Mendeley, Reference Manager are recommended.
References: References must be numbered in order of appearance in the text (including table captions and figure legends) and listed individually at the end of the manuscript. We recommend preparing the references with a bibliography software package, such as EndNote, ReferenceManager or Zotero to avoid typing mistakes and duplicated references. We encourage citations to data, computer code and other citable research material. If available online, you may use reference style 9. below.